# Evaluation of Sleep-Disordered Breathing and Respiratory Dysfunction in Children with Myotonic Dystrophy Type 1—A Retrospective Cross-Sectional Study

**DOI:** 10.3390/biomedicines13040966

**Published:** 2025-04-15

**Authors:** Mihail Basa, Jovan Pesovic, Dusanka Savic-Pavicevic, Stojan Peric, Giovanni Meola, Alessandro Amaddeo, Gordana Kovacevic, Slavica Ostojic, Aleksandar Sovtic

**Affiliations:** 1Department of Pulmology, Mother and Child Health Care Institute of Serbia “Dr. Vukan Cupic”, 11070 Belgrade, Serbia; mihail.basa@imd.org.rs; 2Center for Human Molecular Genetics, Faculty of Biology, University of Belgrade, 11158 Belgrade, Serbia; jovan.pesovic@bio.bg.ac.rs (J.P.); duska@bio.bg.ac.rs (D.S.-P.); 3Faculty of Medicine, University of Belgrade, 11158 Belgrade, Serbia; stojan.peric@med.bg.ac.rs (S.P.); gordana.kovacevic@imd.org.rs (G.K.); slavica.ostojic@imd.org.rs (S.O.); 4Neurology Clinic, University Clinical Center of Serbia, 11158 Belgrade, Serbia; 5Department of Neurorehabilitation Sciences, Casa Di Cura Igea, 20129 Milan, Italy; giovanni.meola@unimi.it; 6Department of Biomedical Sciences for Health, University of Milan, 20133 Milan, Italy; 7Fondazione Malattie Miotoniche ETS—FMM ETS, 20144 Milan, Italy; 8Institute for Maternal and Child Health-IRCCS “Burlo Garofolo”, 34137 Trieste, Italy; alessandro.amaddeo@burlo.trieste.it; 9Neurology Department, Mother and Child Health Care Institute of Serbia “Dr. Vukan Cupic”, 11070 Belgrade, Serbia

**Keywords:** myotonic dystrophy type 1, sleep-disordered breathing, transcutaneous capnometry, sleep study, noninvasive ventilation

## Abstract

**Background/Objectives**: Myotonic dystrophy type 1 (DM1) is a rare neuromuscular disorder characterized by respiratory dysfunction that significantly impacts quality of life and longevity. This study aimed to explore the outcomes of pulmonary function tests and sleep-disordered breathing (SDB) workups in children with DM1 and to identify the factors contributing to SDB. **Methods**: A retrospective study examined patients’ medical records, including genetic analyses, clinical characteristics, and noninvasive pulmonary function testing (PFT), when possible. The Pediatric Sleep Questionnaire (PSQ), arterial blood gases, polygraphy, and overnight transcutaneous capnometry (PtcCO_2_) were used to assess SDB. **Results**: The size of CTG expansion in the *DMPK* gene directly correlated with the severity of respiratory complications and the need for early tracheostomy tube insertion in 7/20 (35%) patients. A total of 13/20 (65%) children were available for respiratory evaluation during spontaneous breathing. While moderate/severe obstructive sleep apnea syndrome (OSAS) and hypoventilation were confirmed in 4/13 (31%) children, none of the patients had mixed or dominantly central sleep apnea syndrome. There was no correlation between apnea–hypopnea index (AHI) or PtcCO_2_ and the presence of SDB-related symptoms or the PSQ score. Although a significant correlation between AHI and PtcCO_2_ was not confirmed (*p* = 0.447), the oxygen desaturation index directly correlated with PtcCO_2_ (*p* = 0.014). **Conclusions**: While SDB symptoms in children with DM1 may not fully correlate with observed respiratory events or impaired gas exchange during sleep, a comprehensive screening for SDB should be considered for all patients with DM1. Further research into disease-specific recommendations encompassing the standardization of PFT, as well as overnight polygraphic and capnometry recordings, could help to guide timely, personalized treatment.

## 1. Introduction

Myotonic dystrophy type 1 (DM1) is a rare inherited neuromuscular disease (NMD) that can severely affect quality of life and life expectancy [1]. An autosomal-dominantly inherited trinucleotide cytosine–thymine–guanine (CTG) repeat expansion in the non-coding part of the myotonic dystrophy protein kinase (*DPMK*) gene on the long arm of chromosome 19 causes the disease [1,2]. Muscle weakness and myotonia, along with multiorgan involvement (the heart, eyes, central nervous system, and gastrointestinal and endocrine system), are a hallmark of the disease. However, respiratory dysfunction along with cardiac involvement remains the primary factor contributing to a reduced lifespan and a significant factor affecting the quality of life [3,4]. Even though there have been big steps forward in diagnosing and treating respiratory problems in DM1 and other NMDs, the cause of sleep-disordered breathing (SDB) is still not fully clear [3,4].

The etiology of respiratory complications is multifactorial and still partly elusive. Impaired central breathing control, an obstructive ventilatory pattern (craniofacial abnormalities, enlarged adenotonsillar tissue, and pharyngomalacia), and ventilatory incapacity caused by muscle weakness, chest wall deformities, and scoliosis combine to develop respiratory complications [3,4,5]. Respiratory problems are more common in the congenital form of DM1 than in childhood and juvenile forms of the disease [6]. However, the longitudinal changes in pulmonary functions, blood gas analyses, and polysomnographic findings during childhood remain unclear [3,7]. Although video-surveilled polysomnography (PSG) remains the gold standard for SDB, doubtful adherence and possible disturbance during recordings have urged the introduction of clinical tools that offer better comfort. A method that combines questionnaire scores, pulmonary function tests, overnight polygraphy, and noninvasive transcutaneous blood gas measurement is possible and gives reliable results [8,9]. The identification of potential additional factors that may contribute to the genesis of SDB-related symptoms is crucial, particularly in patients with DM1 who have normal overnight polygraphy and gas exchange parameters [9].

Published studies have extensively reported sleep disturbance diagnostics and ventilatory modalities in adults diagnosed with DM1 [10,11]. The lack of data from the pediatric population is mainly due to the limited number of studies [3,12]. Given the significant differences in disease progression, comorbidities, and severity between children and adults, simply extrapolating data from adult studies and applying them to the pediatric population may not fully address all of the challenges associated with DM1 in children [12,13]. Notably, the lack of reports from middle-income countries is of particular significance. In environments with limited resources, caregivers and parents meet challenges such as limited access to diagnostic tools and the imposition of essential respiratory management restrictions [14,15]. These restrictions include the prescription of ventilation equipment according to the patient’s particular need and the excessive cost of respiratory clearance and cough-assisted devices, which are not reimbursed by national health care services [14,15]. People with NMD typically employ regular respiratory secretion clearing techniques, which, either alone or in conjunction with ventilatory support, constitute a crucial tool for overall pulmonary health care [16]. We still need to clarify the precise function and standards for its effective role in DM1 and to reap its benefits.

This study aimed to elucidate the experience of two pediatric centers regarding the outcomes of pulmonary function tests and SDB workup in children diagnosed with DM1. Secondarily, the aim was to spot and describe the features potentially contributing to SDB and SDB-related symptoms, and to highlight the differences in comparison to the results of studies including adults with DM1. Additionally, we focused on the function of the ventilator’s built-in software analysis in children reliant on long-term mechanical ventilation (MV) at home for whom sleep studies were unsuitable.

## 2. Materials and Methods

### 2.1. Study Design

The retrospective study encompassed the medical records of children diagnosed with DM1 from 2016 to 2025 at the following two pediatric centers: the Institute for Mother and Child Health Care of Serbia (Belgrade, Serbia) and the Institute for Maternal and Child Health-IRCCS “Burlo Garofolo” (Trieste, Italy). The Department of Neurology performed a thorough neurological evaluation of all patients, while the Departments of Pulmonology conducted respiratory system assessments. Genetic tests were performed at the Center for Human Molecular Genetics, University of Belgrade—Faculty of Biology. The genetic test for one patient was performed in a center in Trieste, Italy. The study included individuals who regularly attended clinical visits and outpatient follow-ups, and who were eligible for sleep studies and continuous capnometry. The study excluded those with incomplete medical records or who were unwilling to participate in a respiratory assessment. Each patient’s parents provided written consent for their participation in the study. The study was conducted in accordance with the Declaration of Helsinki and approved by the Institutional Ethics Committee of Mother and Child Health Care Institute of Serbia (protocol code 8/4; date of approval: 20 January 2025).

### 2.2. Genetic and Neuromuscular Assessment

Individuals were classified into the following three clinical forms according to the age of symptom onset: congenital, which begins within the first year of life; childhood, which begins between 13 months and 10 years of age; juvenile, which manifests clinically between the ages of 11 and 20 [17]. Clinical diagnosis was confirmed genetically using repeat-primed polymerase chain reaction (PCR), which provides information on the presence of expansion only [18]. However, the *DMPK* expansion size data were available for 9 selected patients [1]. For the remaining 11 individuals, the retrospective determination of the expansion size was performed using small-pool PCR analysis, since their diagnostic DNA samples were of good quality [19].

Data on the patients encompassed demographics, pregnancy details, fetal movements, prenatal anomalies (polyhydramnios and fetal ventriculomegaly), and birth information. We documented the grade of hypoxic–ischemic encephalopathy (HIE), respiratory abnormalities, feeding difficulties, and cardiac involvement. Brain magnetic resonance imaging (MRI) was a standard part of the diagnostic workup and was available for each patient. Clinical psychologists evaluated psychomotor development with the Serbian adaptation of the Wechsler Intelligence Scale for Children (WISC).

### 2.3. Respiratory Assessment

The first step of the respiratory assessment involved a physical examination and, if possible, noninvasive pulmonary function testing. The following phase involved a comprehensive evaluation of sleep-disordered breathing, incorporating arterial blood gas analysis, the Pediatric Sleep Questionnaire (PSQ), and night-time transcutaneous capnometry (PtcCO_2_) alongside polygraphy [20,21].

### 2.4. Pulmonary Function Testing

Children who were cooperative and able to perform specific maneuvers underwent clinical tests of respiratory function, including spirometry, peak cough flow (PCF), maximal inspiratory (MIP) and expiratory (MEP) pressure (as a surrogate for respiratory muscle strength tests), and body plethysmography in each case. Suitability for executing maneuvers required for pulmonary function tests (PFTs) was assessed on an individual basis and in accordance with the preceding psychological evaluation. Although the protocol design favored data quality, it had limits, because the majority of participants were unable to make a sustained effort. Considering the lack of internationally accepted and validated PFT values for children diagnosed with DM1, the following established norms derived from other NMDs were used: PCF < 160 L/min, MIP < −60 cm H_2_O, total lung capacity (TLC) < 70% predicted, and FVC < 40% predicted, signifying markedly compromised parameters. We took each measurement either in the sitting position or upright position. Technical obstacles prevented us from taking any measurements in a supine/lying position, and comparing possible differences in the obtained values that indicated altered diaphragm and chest wall mechanics. Inefficient cough reflex (documented by a decreased PCF in patients where measurement was possible) led to the introduction of the airway clearance mechanical devices (Cough Assist^®^ or Comfort Cough^®^). Cough augmentation techniques in tracheostomized children, and those unable to perform PFT due to intellectual disability, were used as well.

### 2.5. Nocturnal Gas Exchange and Breathing Evaluation

The overnight gas exchange was documented by using PtcCO_2_ (V-STATS, SenTec AD, Therwil, Switzerland), followed by arterial blood gas analyses upon awakening. More than 2% of the total sleep time (TST) with PtcCO_2_ > 50 mmHg was indicative of nocturnal hypoventilation [22]. We conducted a concurrent sleep study (polygraphy) whenever possible in spontaneously breathing children using the Embla MPR (Embla Systems, Broomfield, CO, USA). Two experienced physicians evaluated respiratory events following the American Academy of Sleep Medicine (AASM) [23]. The primary indication for long-term noninvasive ventilation (NIV) was hypercapnic respiratory failure, as shown by increased PtcCO_2_. Nevertheless, we employed tracheostomy and chronic invasive ventilation at home for those with either NIV failure or acute respiratory insufficiency and weaning failure. Each child attended both inpatient and outpatient visits, during which they underwent ventilator in-built software data analysis for daily usage, leaks, residual respiratory events, and the number of patient-triggered breaths. The inspiratory trigger was considered to be sensitive if a value up to 2 was chosen on a 9-point numeric scale. During inpatient visits, each invasively ventilated child underwent noninvasive transcutaneous gas exchange.

### 2.6. Endoscopic Airway Evaluation

Flexible bronchoscopy was a routine part of the evaluation in cases with overnight impaired gas exchange and moderate-to-severe OSAS. All of the cases with congenital forms of disease and inserted tracheostomy tubes had endoscopic examinations before and at least six months after the tracheotomy. This study also included endoscopic examination for children with mild OSAS. The procedure was performed via the nasal route by experienced bronchologists. Each procedure followed informed parental consent. After the evaluation of the upper airways, analgosedation with intravenous midazolam and epimucous lidocaine application preceded subglottic and lower airway assessment.

### 2.7. Statistical Analysis

IBM SPSS Statistics 27 for Windows (IBM Corp. Released 2020. IBM SPSS Statistics for Windows, Version 27.0. Armonk, NY, USA: IBM Corp) was used to analyze the statistical data. Student’s T-test was used to calculate the statistical difference between normally distributed variables, while the Welch t-test was used in case of unequal variances of normally distributed variables. The Mann–Whitney U test was used to analyze the differences in medians between non-normally distributed variables. We assessed the differences between categorical variables using the chi-square test. The Pearson correlation coefficient for normally distributed parameters and Spearman’s rank correlation coefficient for nonparametric data and ordinal variables were used to evaluate the correlations. Point biserial correlation was employed if one variable was dichotomous. We used binomial logistic regression to evaluate the relationship between a binary outcome and multiple predictor variables. Statistically significant differences were documented by *p* < 0.05.

## 3. Results

### 3.1. Patient Overview and General Data

Out of the 26 children with DM1, the parents or caregivers of 6 children (23% of total available number) decided not to participate in the study. These children were omitted from the study group. The study included a total of 20 children diagnosed with DM1—19 patients from the center in Belgrade, Serbia, and 1 child from the center in Trieste, Italy. Those with the congenital form (17/20, 85%) significantly outnumbered those with the childhood (1/20 or 5%) and juvenile type (2/20 or 10%) of the disease (*p* = 0.003). Males were significantly predominant over females (*p* = 0.04). Table 1 shows the demographic data.

The CTG repeat expansion size was statistically significantly larger in children with respiratory insufficiency and a continuous need for invasive respiratory support (1621 repeats; SD ± 280) in comparison to the spontaneously breathing children (1076 repeats; SD ± 472) (*p* = 0.013), with a medium effect size (Cohen’s d = 0.546).

### 3.2. Questionnaire Scores and Pulmonary Function Testing

The most frequently reported symptom characteristic for SDB was snoring (5/13 or 38%), followed by daytime sleepiness and fatigability (2/13 or 15%). However, the majority were asymptomatic. There was no statistical difference in the number of CTG repeats between children with and without symptoms (*p* = 0.229). Accordingly, the PSQ score exceeded the cut-off value of 0.33 in one child, suggesting SDB, while the rest of the children had normal PSQ scores. The DM1 clinical form and intellectual ability strongly correlated with the ability to perform pulmonary function tests (point biserial r = −0.970; *p* < 0.001). Since only one child with a congenital form of disease was able to sustain and acceptably exert the effort required for the PFT, all three children diagnosed with the childhood and juvenile type of the disease were able to successfully complete the measurements. The mean age at the first successful measurement was 12 years. Two children had a normal total lung capacity (TLC > 80% of predicted values) and one had slightly reduced TLC (79% of predicted), referring to a mild restrictive ventilatory pattern. TLC was not available for one child. There were no results suggestive of an obstructive pattern. However, the MIP and MEP were uniformly decreased; the mean MIP was −40 cm H_2_O and the MEP was 29 cm H_2_O. Furthermore, there was a general decrease in the PCF, with an average value of 193 L/min.

### 3.3. Nocturnal Gas Exchange and Breathing Evaluation

Prematurity with significant perinatal complications (asphyxia, HIE, or respiratory distress syndrome) strongly correlated with the necessity for intubation and mechanical ventilation in neonatal age (*p* = 0.017). Notably, six out of seven (86%) chronically invasively ventilated children experienced premature delivery and one of the perinatal complications. Ventilation resulted in gas exchange normalization in each case. Additionally, each patient had in-built software analyzed for daily usage, leaks, residual respiratory events analysis, and the number of patient-triggered breaths.

Out of the 20 participants in the study, 13 children, or 65%, underwent simultaneous sleep studies and overnight transcutaneous capnometry during spontaneous breathing. The average age at the time of recording was 5.4 years (range 4 months–17 years). A total of 4 children (31%) had no SDB expressed through the apnea–hypopnea index (AHI), while 9/13 (69%) had an uneven distribution of increased AHI, typically indicating obstructive patterns of ventilation. A total of 6/13 patients (46%) had mild OSA, 2/13 (15%) had moderate OSA, and one child had severe OSA. None of the patients had mixed or dominantly central sleep apnea syndrome (CA). Table 2 lists the results of the polygraphy recordings.

The AHI was in positive correlation to the oxygen desaturation index (ODI) (Spearman’s rho 0.606; *p* = 0.028), as shown in Figure 1. There was no significant statistical difference in the AHI and ODI between congenital and childhood/juvenile forms of the disease (*p* = 1.0 and *p* = 0.585, respectively). Additionally, we did not find a statistical difference in terms of the AHI or ODI between asymptomatic individuals and those with at least one symptom specific to SDB (*p* = 0.532 and *p* = 0.950, respectively).

Although most of the children had normal PSQ scores, the sole child with a high PSQ score, who experienced an infantile beginning of the disease, also had a severely disordered obstructive breathing pattern during sleep. Children with MRI findings of reduced brain volume had a significantly lower ODI compared to those without MRI brain volume abnormalities (Welch t test = 3.883, *p* = 0.004, with Cohen’s d of 0.758), as shown in Figure 2. There was no correlation between reduced brain volume and other polygraphy or capnometry indices in spontaneously breathing patients, nor did we find an association between BMI and overnight recording parameters (*p* > 0.05).

Among the 13 children who had PtcCO_2_ during spontaneous breathing, 4 fulfilled the criteria for alveolar hypoventilation, including 3 with a congenital form and 1 with a juvenile form of the disease. The mean PtcCO_2_ was 46.70 mmHg (SD 7.8 mmHg)—data from the capnometry are given in Table 2. There was no statistical difference in PtcCO_2_ between the two forms of the disease, nor did we find a correlation between PtcCO_2_ and the presence of SDB-related symptoms (*p* = 0.815 and *p* = 1.0, respectively). In addition, there was no correlation between the overnight sleep parameters and PFT values (TLC, MIP, and MEP) (*p* > 0.05). We did not confirm the correlation between the age of the spontaneously breathing patients and the results of the polygraphic or capnometry parameters. There was no significant correlation between the AHI and PtcCO_2_ (*p* = 0.447), but the analysis showed a strong positive correlation between the ODI and PtcCO_2_ (Pearson’s r = 0.64; *p* = 0.014), as shown in Figure 3. We did not find a significant correlation between the PtcCO_2_ and mean heart rate during sleep (*p* = 0.749), nor between the AHI or ODI and mean heart rate (*p* = 0.184 and *p* = 0.164, respectively).

The number of CTG repeats did not correlate with the overnight sleep parameters in spontaneously breathing individuals—AHI, ODI, PtcCO_2_, and mean SpO_2_ (*p* = 0.241, *p* = 0.389, *p* = 0.432, and *p* = 0.830, respectively).

### 3.4. Bronchoscopic Evaluation

Among the five children with mild SDB, only one had pharyngomalacia. Pharyngomalacia was observed in two out of three children who met the criteria for NIV and in each case with an inserted tracheostomy tube (seven out of seven), which was significantly more frequent than in the others without moderate/severe SDB or alveolar hypoventilation (*p* = 0.007). Adenotonsillar tissue was mildly enlarged in one NIV candidate. Neither of the spontaneously breathing participants had significant lower airway structural anomalies.

### 3.5. Respiratory Management and Follow-Up

Long-term mechanical ventilation at home started in a total of 11/20 children (55%). Due to global respiratory insufficiency and the inability to wean from conventional MV, seven children underwent tracheostomy in infancy. In-built software insight uniformly showed excellent adherence to the ventilation. The mortality rate in our study group stems entirely from a cohort of tracheostomized patients—six out of seven (86%) of the reported death outcomes resulted from acute deterioration of chronic respiratory failure.

Among the children who underwent a thorough respiratory evaluation during spontaneous breathing, four showed nocturnal alveolar hypoventilation, and noninvasive ventilation during sleep was the preferred treatment. Three patients had good adherence according to the built-in software data analysis (NIV was in charge for >90% of the nights for an average period of 7 h), while the adherence to the treatment was not satisfactory in the fourth case, despite the initial positive effects on nocturnal alveolar gas exchange (<20% of the nights above 4 h of continuous usage).

The devices for mechanically assisted cough were typically recommended for all mechanically ventilated patients (both invasively and noninvasively) and those with low peak cough flow values.

## 4. Discussion

Our study presents further data on the respiratory assessment in a group of children with DM1, revealing diverse outcomes in SDB assessments and emphasizing that accurate workup, monitoring, and therapy are attainable in resource-constrained settings. Given that respiratory complications are the primary cause of shortened lifespan in individuals with DM1, a respiratory evaluation is one of the recommended diagnostic steps within the diagnostic follow-up [16]. While the strategy of prioritizing the early evaluation of SDB is useful, there remain concerns on patient selection and the specific age at which thorough investigations should be started. Regarding the lack of correlation between age and the recorded SDB/alveolar hypoventilation in our study, we advocate for an “as-early-as-possible” approach to SDB screening until future investigations establish widely recognized guidelines. Unlike the conventional approach for NMD, where SDB is more probable for specific PSQ and PFT thresholds, the scope of SDB screening in DM1 should be broadened for every child with DM1 regardless of the questionnaire and noninvasive pulmonary function testing outcomes, as our study revealed no correlation between these variables and overnight recordings.

The lack of correlation between symptoms, such as excessive daytime sleepiness or snoring, and the sleep study and capnometry has prompted the need for alternative approaches that can identify individuals who need an as-early-as-possible SDB evaluation [11,13]. Understanding the specifics of the disease itself and prompt overnight gas exchange assessment are crucial for effective treatment. The PFT yielded respectable results in other NMDs, with certain predictive values strongly highlighting patients prone to SDB [24]. Contrary to that, our results confirmed that frequent intellectual disability interferes with the possibility of obtaining both dependable PFT maneuvers and results in children with DM1 [3,25]. Furthermore, even though the small sample size limits a reliable conclusion; among those capable of performing PFT in our study, SDB was not consistent with reduced respiratory muscle strength, which was consistent to the findings of other authors [3,25,26]. Given the difficulties with performing spirometry, forced oscillometry may become a viable alternative for individuals incapable of performing standard spirometry [8]. Although disease-specific values are still unreported, research has demonstrated higher positive predictive values of forced oscillometry parameters compared to FVC ≤ 60% [8].

The overall phenotypic severity principally depends on the DM1 disease form and the genotype [1]. Our results confirm that the most extreme respiratory consequences—chronic global respiratory insufficiency that makes a child dependent on artificial ventilation in first weeks of life—directly correlate with the higher number of CTG repeats. In particular, problems with the central control of breathing can further deteriorate with severe neurologic dysfunction following perinatal asphyxia and serious perinatal neuroradiologic complications [1,27]. Afterwards, severe generalized muscle weakness is among the most striking features in affected individuals, suggesting a propensity for both obstructive and restrictive ventilatory disorders [1,28].

Although the etiology of the respiratory manifestations in spontaneously ventilated patients is likely multifactorial, obstructive etiology has been strongly confirmed in our study [1,25,26]. The correlation between SDB symptoms, polygraphy results, and overnight blood gas analyses with the number of CTG repeats is less obvious in children without a need for permanent invasive ventilation from infancy—conflicting results have emanated from multiple studies [5,26,29]. Our results did not confirm the correlation between genotype and polygraphic/capnometry parameters in the evaluated cases. The lack of evidence advocating for a relationship between the genotype- and SDB-related parameters suggests that possible ancillary-acquired mechanisms influence the regulation of breathing during sleep—notably, the natural slow progression of the disease with years and the associated impact of the body mass index [5,30]. While obesity solely added to the risk for SDB in adults with DM1, there were no obese children in our study group [30]. Speculation that underweights in terms of decreased total body mass index, as a contributor to muscle weakness, can be a potential contributor to disturbed overnight gas exchange was not confirmed. Although the detrimental effects of low BMI on muscle strength and SDB are known in other NMDs, and an association between underweights and OSA is described in non-NMD children, that has not been confirmed in children with DM1 so far [31,32]. While endoscopic airway evaluation is acknowledged as a component of the assessment for SDB associated to NMD, to our knowledge, children with DM1 have not undergone systemic bronchologic evaluation. Considering the common occurrence of pharyngomalacia in NMD, the upper airway pathology should not be underestimated as a potential contributing factor to SDB in DM1. Our findings support the usage of diagnostic tools for an upper airway assessment in conditions complicated by OSAS, as the results may influence the long-term treatment [33]. Given the significant prevalence of pharyngomalacia shown in our study among individuals necessitating ventilatory support, a comprehensive endoscopic assessment of the upper airway in children with DM1 and confirmed SDB may be prudent.

Principally, the findings supporting the hypothesis about the central origin of excessive somnolence and disturbed sleep architecture in adults with DM1 may play an important role for future investigations [11]. While the longitudinal relation between the MRI brain findings and PSG findings has not been well scrutinized yet in children and adults, evidence about the central contribution to SDB seems reasonable—positive correlation was confirmed between the MRI brain volumetric findings and ODI in spontaneously ventilated subjects in our study. In addition, in our study group, invasively ventilated children, those most severely affected, had the most pronounced volumetric white matter changes in the CNS. Because they were not appropriate for sleep studies during spontaneous breathing, and thus not included in this section of the statistical analysis, these data should be interpreted with caution. Noteworthy, a possible explanation for a lack of correlation of the volumetric brain changes to other polygraphic parameters in this study could come from the cross-sectional design of the study or, more probably, the relatively small number of participants included. The precise mechanism by which the CNS influences sleep-disordered breathing remains less clear, although complex anatomic and biochemical neuronal changes were documented [11,34]. Except for the primary impact on the psychosocial and emotional aspects, the possible mechanism could include altered central control of respiratory muscle activity during sleep, although the central pattern of SDB was not evidenced in our study, and this hypothesis remains speculative [34].

Overall, our results strongly emphasize the need for the inclusion of more patients, and support the introduction of studies designed to involve and track the longitudinal relation between neuroanatomic or neuropathologic CNS changes and PSG findings. Careful longitudinally designed observations would offer better insights into long-term respiratory function and complex cerebral–muscular–respiratory axis interactions during sleep [11,34]. Hitherto, the slow progression of respiratory disease and stable overnight PtcCO_2_ during follow-up in children with DM1 suggests the need for the introduction of more sensitive capnometry thresholds specific to this population [3,25,26,35].

The limited cooperation of patients with neurodevelopmental disorders, including those diagnosed with DM1 in regard to both the diagnostic workup and ventilatory treatment, imposes significant concern on clinical grounds [16,35,36]. Our study encountered comparable difficulties, as evidenced by the number of families that declined respiratory evaluation. The necessity for implementing a respiratory workup designed to enhance patient comfort has been emphasized—SpO_2_ and PtcCO_2_ monitoring has become a widely used tool that significantly facilitates patient comfort and may be useful in the diagnosis of severe OSA in children [37]. While the assessment of SDB by pulse oximetry trends do not exceed that of the attended PSG, our findings on the correlation between the ODI and both the AHI and PtcCO_2_ underscore the significance of the ODI among the crucial screening parameters for SDB. Given the established utility of trend oximetry in resource-limited settings, it may serve as a valuable tool for patients who are noncompliant with the meticulous and time-consuming conventional SDB assessment, and all through the initial evaluation and subsequent follow-up after the initiation of NIV [37,38].

The built-in software of each ventilator is mostly used to check for daily compliance, leaks, and the overall ventilation effectiveness [39,40]. Although the adherence to NIV is difficult for some patients with DM1, partly due to concomitant intellectual disability, this treatment modality has overall satisfactory results in terms of the normalization of gas exchange [3,16].

Despite efforts to collect as much quality data as possible, this study has some limitations. The retrospective cross-sectional study design precluded the monitoring of longitudinal changes associated with sleep-disordered breathing and its resultant multisystemic effects. Furthermore, the reliance on retrospective anamnestic data extracted from medical records bears a risk of revealing an insufficient comprehension of the actual patient health state, which, alongside the potential selection bias from the exclusion of six children whose parents’ declined participation in the study, may compromise the accuracy of the study outcomes. The retrospective design resulted in a genetic analysis over various time periods, although it did not affect the reliability of the genotype–phenotype correlations. Moreover, the relatively low adherence to the respiratory workup, including the SDB evaluation, resulted in a proportionally small study group. Moreover, the available sleep study data included only spontaneously breathing children. Specifically, children with the most severe phenotype of invasive mechanical ventilation at home were unsuitable for the sleep study, making the reported AHI and other sleep study parameters likely understated. Although the analysis of the in-built software allows for an insight into the airflow patterns and the number of patient-triggered breaths, it has not been part of standard diagnostic tools in children with DM1, and those results should be accepted with caution. Furthermore, we performed overnight polygraphy instead of full polysomnography. While polysomnography provides reliable data on the sleep architecture, it has downsides regarding a patient’s comfort, and could have further impeded the adherence to the overnight recordings, especially in children with intellectual disabilities. Finally, the number of children capable of performing PFT was low—those patients were exclusively diagnosed with childhood/juvenile DM1. The disparity in the number of children with congenital and childhood/juvenile DM1 necessitates further examination, since the statistical differences in outcomes between these two groups warrant the inclusion of additional cases.

## 5. Conclusions

SDB remains an important concern in children diagnosed with DM1. Although SDB-related symptoms are not fully explained by respiratory events and disturbed gas exchange during sleep, systematic screening for SDB is recommended in each patient. Muscular weakness in DM1 is slowly progressive, and the referent values of the PFT and symptom scoring systems imported from other NMDs infrequently highlight the point from which SDB is highly possible in DM1, so overnight poly(somno)graphy and capnometry should be performed regularly, after the diagnosis is confirmed. Despite the inconsistent outcomes of polygraphic recordings in our patients, the significance of systemic screening can be justified in the following two respects: it solely indicates potential SDB in DM1, irrespective of other clinical and functional indicators, and, furthermore, adherence with the recommended respiratory support leads to the normalization of abnormal gas exchange.

## 6. Future Directions

Since most pediatric-based studies include a relatively small number of participants, an approach with a multicentric study design could provide more reliable data. Moreover, data unification and international registry-based studies could ease and upgrade the overall diagnosis and therapeutic management of children with DM1. Longitudinally designed studies are essential therewith. Besides a better understanding of the trends related to the PFT and SDB evaluation, it could be the basis for an identification of further genotype–phenotype implications, and both respiratory and non-respiratory determinants of SDB-related symptoms that cannot be explained by PSG or overnight gas exchange disordered parameters solely. Eventually, the creation of disease-specific recommendations based on the standardization of the PFT and overnight polygraphic and capnometry values would generate a framework inside which timely, individually based management could become possible. Finally, the implementation of artificial intelligence in the field of sleep medicine, with a minimalistic burden on patients in terms of the equipment, has been in focus in recent years, and DM1 could benefit from new artificial intelligence-based technologies [37].

## Figures and Tables

**Figure 1 biomedicines-13-00966-f001:**
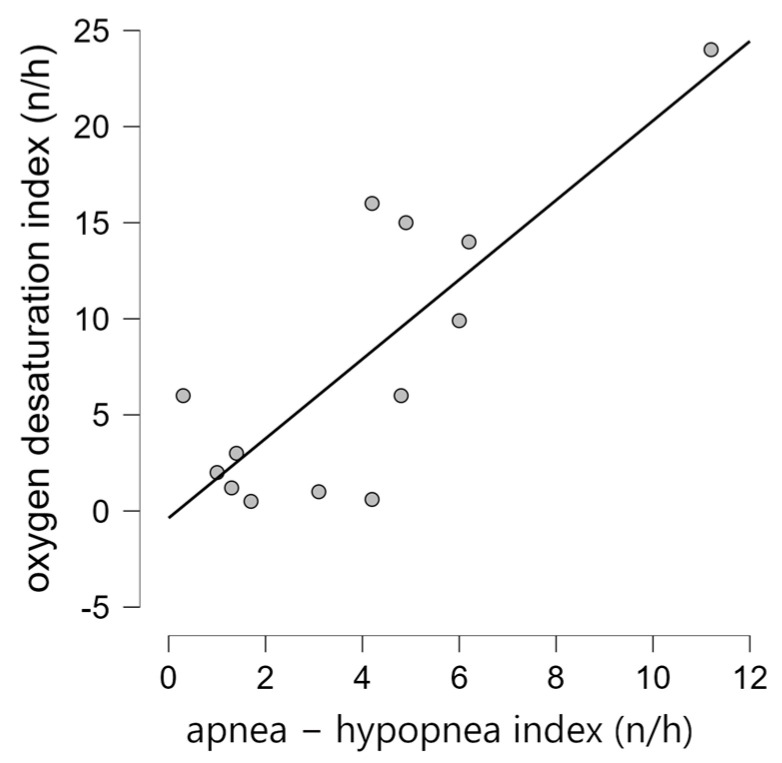
There was a strong positive correlation between the apnea–hypopnea index and oxygen desaturation index.

**Figure 2 biomedicines-13-00966-f002:**
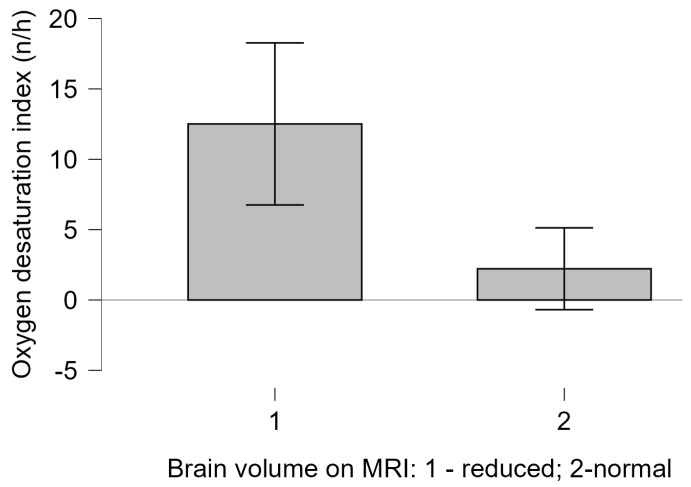
Children with MRI findings of reduced brain volume had significantly higher ODI (mean 12.51; SD 6.88) compared to those without MRI brain volume abnormalities (mean 2.22; SD 2.34).

**Figure 3 biomedicines-13-00966-f003:**
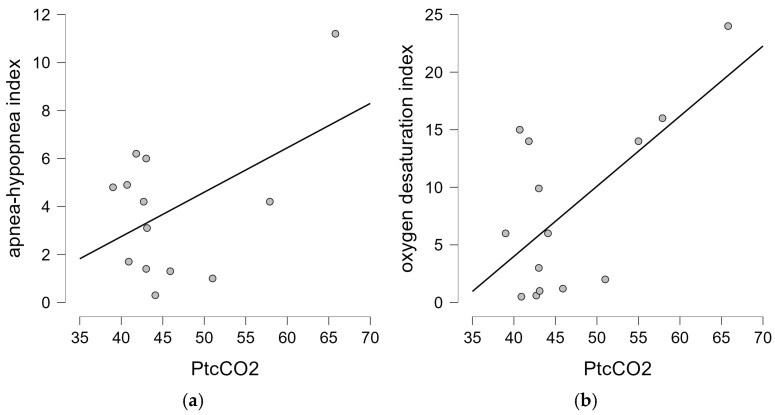
Correlation between the PtcCO_2_ and polygraphic parameters (AHI and ODI): (**a**) there was no significant correlation between the PtcCO_2_ and AHI (*p* = 0.447); (**b**) there was a positive correlation between the PtcCO_2_ and ODI (*p* = 0.014).

**Table 1 biomedicines-13-00966-t001:** Overall characteristics of the study group.

Children with Respiratory Evaluation	20
Successful sleep study and capnometry	13/20 (65%)
Tracheostomy tube in infancy	7/20 (35%)
Age at onset of symptoms	1 month (1–9 months) ^1^ (1 month–12 years) ^2^
Age at diagnosis	2 months (1–36 months) ^1^ (1 month–17 years) ^2^
Age at overnight recordings	5.4 (0.3–17) ^3^
Number of CTG repeats ^2^	1277 (220–1950)
Mean IQ	44.5 (20–80)
Decreased brain volume on MRI	8/13 (61%)
Underweight children (Z score < −2 SD)	5/13 (38%)
Structural cardiac involvement	10/13 (77%) ^4^
Moderate/severe scoliosis	1/13 (7.6%)

^1^—median (interquartile range); ^2^—range; ^3^—mean (range); ^4^—exclusively minor cardiac structural abnormalities (ASD, VSD, and patent ductus arteriosus).

**Table 2 biomedicines-13-00966-t002:** Nocturnal gas exchange findings.

	MEDIAN (IQR)	MIN–MAX
AHI (n/h)	4.2 (1.4–4.9)	0.3–11.2
CAI (n/h)	0.4 (0.1–1)	(0–1.4)
PtcCO_2_ (mm Hg)	43.05 (42–49.7)	39–65.8
Min PtcCO_2_ (mm Hg)	39.9 (33.7–42.9)	28–57.6
Min–max PtcCO_2_ difference	9.1 (6.8–16.5)	6.0–25.9
Time with PtcCO_2_ > 50 mmHg (%)	93.7 (SD ± 5.8)	87–100
Mean SpO_2_ (%)	96.9 (95–98)	91–99
Minimal SpO_2_	87 (86–91.7)	72–94
ODI	6 (1.4–14)	0.5–24
Mean heart rate (n/min) ^1^	98 (SD ± 23.1)	51–135

AHI—apnea–hypopnea index; CAI—central apnea index; TST—total sleep time; ODI—oxygen desaturation index; ^1^—mean value (standard deviation, SD).

## Data Availability

The original contributions presented in this study are included in the article. Further inquiries can be directed to the corresponding author.

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
