# Peer review of "Evaluation of Sleep-Disordered Breathing and Respiratory Dysfunction in Children with Myotonic Dystrophy Type 1—A Retrospective Cross-Sectional Study"

_biomedicines, 2025, doi:10.3390/biomedicines13040966_

Round 1
Reviewer 1 Report
Comments and Suggestions for Authors
Dear Authors
The paper is interesting even if it needs some improvementes, as follow:
- in the title it could be useful to insert the type of study
- no mention about written parental consent, so clarify and insert it in the manuscript
- make the tables more sintetic and modify them as schemes, too much written test is in the tables is not good
- in the Discussion it could be useful to compare these data with other rare disease breathing problems (such as Marfan syndrome) already published
- some references about rare diseases are too old, consider to read and insert more recent ones
- Obstructive sleep apnea in children with Marfan syndrome: Relationships between three-dimensional palatal morphology and apnea-hypopnea index
International Journal of Pediatric Otorhinolaryngology 2018; 112:6-9
- 3D evaluation of maxillary morphology in Marfan growing subjects: a controlled clinical study. Progress in Orthodontics 2019; 20-1-12
-
Obstructive Sleep Apnea, palatal morphology, and aortic dilatation in Marfan syndrome growing subjects: a retrospective study.
Int J Environ Res Public Health. 2021 Mar 16;18(6):3045. doi: 10.3390/ijerph18063045
English can be improved in some sentences
Author Response
Thank you very much for taking the time to review the manuscript. Please find the detailed responses below and the corresponding corrections in track changes in the re-submitted file.
The suggestion 1: in the Discussion it could be useful to compare these data with other rare disease breathing problems (such as Marfan syndrome) already published. Some references about rare diseases are too old, consider to read and insert more recent ones
- Obstructive sleep apnea in children with Marfan syndrome: Relationships between three-dimensional palatal morphology and apnea-hypopnea index, International Journal of Pediatric Otorhinolaryngology 2018; 112:6-9
-3D evaluation of maxillary morphology in Marfan growing subjects: a controlled clinical study. Progress in Orthodontics 2019; 20-1-12
-Obstructive Sleep Apnea, palatal morphology, and aortic dilatation in Marfan syndrome growing subjects: a retrospective study. Int J Environ Res Public Health. 2021 Mar 16;18(6):3045. doi: 10.3390/ijerph18063045
The answer 1: Although addressing breathing problems in Marfan syndrome proved challenging, we included the citation you suggested (the third one) and highlighted the importance of an upper airway evaluation in all individuals with documented obstructive sleep apnea syndrome, particularly taking into account high prevalence of pharyngomalacia in children on NIV from study group. The following statement was rephrased and integrated into the Discussion (lines 392-398):
Considering the common occurrence of pharyngomalacia in NMD, upper airway pathology should not be underestimated as a potential contributing factor to SDB in DM1. Our findings support the usage of diagnostic tools for upper airway assessment in conditions complicated by OSAS as findings may influence long-term treatment. Given the significant prevalence of pharyngomalacia shown in our study among individuals necessitating ventilatory support, a comprehensive endoscopic assessment of the upper airways in children with DM1 and confirmed SDB may be prudent.
The suggestion 2: in the title it could be useful to insert the type of study
The answer 2: We made the suggested corrections by incorporating the retrospective cross-sectional study. With this adjustment, the title of the paper is now: Evaluation of Sleep-Disordered Breathing and Respiratory Dysfunction in Children with Myotonic Dystrophy Type 1 – the retrospective cross-sectional study.
The suggestion 3: no mention about written parental consent, so clarify and insert it in the manuscript
The answer 3: Actually, the statement regarding parental consent is provided in the Materials and Methods section, paragraph 2.1 (Study Design). Please refer to lines 107-108: ’Each patient’s parent provided consent for their participation in the study’. Additionally, there is an informed consent statement at the end of the text (lines 519-521): Informed consent was obtained from parents of all subjects involved in the study. Written informed consent has been obtained from parents of the patients to publish this paper.
Nevertheless, we clarified this further by adding the word 'written' to the following sentence: ’Each patient’s parent provided written consent for their participation in the study’.
The suggestion 4: make the tables more synthetic and modify them as schemes, too much written text is in the tables, is not good
The answer 4: Thank you for ponting this out. A few changes have been made to Table 1. In addition to removing unnecessary text, the row 'adherence to respiratory evaluation in spontaneously breathing patients' has been deleted (the 4th row in the initial version of Table 1), as this point is already addressed in the text. Along with adjustments to the table's dimensions and layout, and some modifications to Table 2, it (hopefully) appears more refined and polished now.
Reviewer 2 Report
Comments and Suggestions for Authors
This study raises important questions about sleep-disordered breathing in children with DM1, but its impact is unclear. Similar studies with larger or comparable sample sizes already exist, yet the authors do not adequately compare their findings or clarify what new insights this study offers.
With only 20 children included, generalizability is limited, and the exclusion of six participants due to caregiver refusal introduces potential selection bias. The reliance on retrospective medical records also raises concerns about data accuracy, which the authors do not address. Additionally, genetic testing was performed at different centers using varying methods, making genotype-phenotype correlations less reliable.
The study acknowledges that only a minority of children could perform pulmonary function tests, yet conclusions about respiratory function are still drawn. Correlations are presented without considering confounding factors, and statistical significance is weak in some key comparisons (e.g., p = 0.013), requiring cautious interpretation.
Most importantly, the authors do not explain how their findings should influence clinical practice. The recommendation for universal screening of DM1 children for sleep-disordered breathing lacks strong evidence, given that only 13 out of 20 underwent sleep studies with mixed results.
Overall, the study needs a clearer discussion of its novelty, stronger statistical analysis, and a more practical interpretation of its findings.
Author Response
We are truly thankful for your detailed review and the time you took to provide us with such helpful feedback. We have incorporated your suggestions, and all revisions are highlighted in track changes.
The suggestion 1: This study raises important questions about sleep-disordered breathing in children with DM1, but its impact is unclear. Similar studies with larger or comparable sample sizes already exist, yet the authors do not adequately compare their findings or clarify what new insights this study offers.
The answer 1: Thank you for this observation! In response, we have made the following changes:
While acknowledging the existence of similar studies, our goal was to share our experience with the diagnostic aspects of pulmonary involvement in children with DM1, particularly as this study presents data from a center with limited resources. We addressed this in the added lines 334-336:
Our study presents further data on respiratory assessment in a group of children with DM1, revealing diverse outcomes in SDB assessments and emphasizing that accurate work up, monitoring, and therapy are attainable in resource-constrained settings.
In line with this, we emphasize the importance of the oxygen desaturation index as a non-invasive and easy-to-tolerate tool. In our study, it showed a correlation with both the apnea-hypopnea index and PtcCO2, making it particularly valuable in resource-limited settings, as well as in managing the often challenging and noncompliant patients with DM1 and their families. Thus, we added lines 433-439 (please, see this paragraph as a part of the answer on the suggestion below about how our findings should influence clinical practice).
Finally, we emphasized the role of flexible bronchoscopy as part of the systemic evaluation for all patients with documented SDB. As far as we know, endoscopic evaluation was not reported as a part of systemic evaluation of children with SDB in DM1. Our study revealed a significantly higher frequency of pharyngomalacia in those requiring NIV (lines 392-398):
Considering the common occurrence of pharyngomalacia in NMD, upper airway pathology should not be underestimated as a potential contributing factor to SDB in DM1. Our findings support the usage of diagnostic tools for upper airway assessment in conditions complicated by OSAS as findings may influence long-term treatment. Given the significant prevalence of pharyngomalacia shown in our study among individuals necessitating ventilatory support, a comprehensive endoscopic assessment of the upper airways in children with DM1 and confirmed SDB may be prudent.
The suggestion 2: With only 20 children included, generalizability is limited, and the exclusion of six participants due to caregiver refusal introduces potential selection bias. The reliance on retrospective medical records also raises concerns about data accuracy, which the authors do not address.
The answer 2: Given the retrospective design of the study, we fully acknowledge the potential limitations. These concerns are addressed in the final paragraph of the Discussion section, where the study's limitations are now incorporated. Specifically, we have added the following sentences (lines 446-452):
The retrospective cross-sectional study design precludes the monitoring of longitudinal changes associated with sleep-disordered breathing and its resultant multisystemic effects. Furthermore, reliance on retrospective anamnestic data extracted from medical records bears a risk of revealing an insufficient comprehension of actual patient health state, which alongside potential selection bias from the exclusion of six children whose parents’ declined participation in the study, may compromise the accuracy of study outcomes.
Considering that MD1 is a rare disease, we highlighted the importance of multicentric studies to provide deeper insights into this topic (lines 484-485).
The suggestion 3: Additionally, genetic testing was performed at different centers using varying methods, making genotype-phenotype correlations less reliable.
The answer 3: In fact, as mentioned in the text, genetic testing for 19 patients was conducted at one center, while genetic testing for only one patient was performed at the second center. Please refer to lines 102-104 for further details:
Genetic tests were performed at the Center for Human Molecular Genetics, University of Belgrade – Faculty of Biology. Genetic test for one patient was done in center in Trieste, Italy.
Additionally, in the Results section, we confirmed that 19 children were diagnosed and treated at one center, while only one child was diagnosed and treated at the second center (lines 203-205).
The use of two techniques (Repeat-primed PCR and small-pool PCR) for genetic analyses can be viewed as a consequence of the retrospective design of the study (It covers a timeframe of about ten years), but may be considered a limitation, although it doesn't affect the results of the genetic testing in any manner. Therefore, we have highlighted this point in the Study Limitations paragraph by adding the following statement (lines 452-454):
The retrospective design resulted in the utilization of two distinct techniques for genetic analysis (Repeat-primed PCR and small-pool PCR) over various time periods, although it did not affect the reliability of genotype-phenotype correlations.
The suggestion 4: The study acknowledges that only a minority of children could perform pulmonary function tests, yet conclusions about respiratory function are still drawn.
The answer 4: With only three children able to undergo pulmonary function tests, it seemed superfluous to expand the discussion or make recommendations based on these limited findings. Nevertheless, we addressed the role of PFT in DM1 (lines 353-359) and included an additional statement in the Conclusions section (lines 475-478). Given that the mean age at the first successful measurement in our study group was 12 years, and the majority of patients were much younger and unable to perform PFT, we also added a statement regarding the potential future role of FOT (Forced Oscillometry Technique). This technique, which can be applied to children as young as 3 years old, offers valuable insights into lung function without requiring significant cooperation from the patient (lines 360-363). This could serve as the focus of future investigations.
Given the difficulties with performing spirometry, forced oscillometry may become a viable alternative for individuals incapable of performing standard spirometry. Although disease-specific values are still unreported, the research demonstrated high positive predictive values of forced oscillometry parameters versus FVC ≤ 60%.
The suggestion 5: Correlations are presented without considering confounding factors, and statistical significance is weak in some key comparisons (e.g., p = 0.013), requiring cautious interpretation.
The answer 5: Thank you for this observation! Since statistically significant differences were determined by a p-value of < 0.05, we followed this threshold for calculating significance. However, recognizing that the p-value alone does not provide much insight into the strength of the correlation or statistical effect, we included additional indicators of effect size (Point-Biserial r, Cohen's d, Spearman’s rho, Pearson’s r) whenever p < 0.05. By providing these effect size indicators, we aim to offer a more standardized measure of the relationship between variables, thereby contributing to a more cautious and nuanced interpretation of the results.
The suggestion 6: Most importantly, the authors do not explain how their findings should influence clinical practice.
The answer 6: Thank you for bringing this to our attention. In light of your suggestion, we have revised some sections to clarify this issue.
To emphasize the impact of these findings on our clinical practice, we revised the first paragraph of the Discussion to highlight two critical points: when to evaluate a patient for SDB and whom to evaluate, particularly in relation to the findings from the questionnaire scoring system (PSQ) and pulmonary function testing (PFT). Accordingly, we have added the following section (lines 339-348):
While the strategy of prioritizing early evaluation of SDB is useful, there remain concerns on patient selection and the specific age at which thorough investigations should be started. Regarding the lack of correlation between age and recorded SDB/alveolar hypoventilation in our study, we advocate for an "as early as possible" approach to SDB screening until future investigations establish widely recognized guidelines. Unlike the conventional approach for NMD, where SDB is more probable for specific PSQ and PFT thresholds, the scope of SDB screening in DM1 should be broadened for every child with DM1 regardless of questionnaire and noninvasive pulmonary function testing outcomes, as our study revealed no correlation between these variables and overnight recordings.
In addition, we included the paragraph emphasizing the importance of the oxygen desaturation index (ODI) as an important screening tool in resource-limited facilities and non-adherent patients, all of which is highly relevant to our everyday practice. We added the following paragraph (lines 433-439):
While the assessment of SDB by pulse oximetry trends don’t exceed that of attended PSG, our findings on the correlation between ODI and AHI, PtcCO2 and MRI indicated reduced brain volume, underscore the significance of ODI among crucial screening parameters for SDB. Given the established utility of trend oximetry in resource-limited settings, it may serve as a valuable tool for patients who are noncompliant with the meticulous and time-consuming conventional SDB assessment, all through the initial evaluation and subsequent follow-up after the initiation of NIV.
The suggestion 7: The recommendation for universal screening of DM1 children for sleep-disordered breathing lacks strong evidence, given that only 13 out of 20 underwent sleep studies with mixed results.
The answer 7: To underscore the importance of systemic SDB screening in children with DM1, we revised the conclusion and added the following statement (lines 478-482):
Despite the inconsistent outcomes of polygraphic recordings in our patients, the significance of systemic screening can be justified in two respects: it solely indicates potential SDB in DM1, irrespective of other clinical and functional indicators, and furthermore, adherence with the recommended respiratory support leads to the normalization of abnormal gas exchange.
The suggestion 8: Overall, the study needs a clearer discussion of its novelty, stronger statistical analysis, and a more practical interpretation of its findings.
The answer 8: Once again, we sincerely appreciate the time and effort you have dedicated to reviewing this manuscript. We hope that the changes we have made address your valuable feedback and offer a more comprehensive and practical approach to the topic discussed in the paper. With the revisions to the statistical analyses—including the addition of new graphs and statistical parameters that better indicate the strength of correlations and statistical effects—along with a more thorough review of the practical aspects, the paper is now more complete and robust overall.
Reviewer 3 Report
Comments and Suggestions for Authors
Basa et al. performed a study to evaluate sleep disorders in patients with myotonic dystrophy type 1. My comments are:
Please provide a complete description of all abbreviations at the first presentation, including the software description. E.g. IBM SPSS , please provide city of development and full description.
How were the variables distributed? Provide the Spearman correlation graphs as supplementary material for data validation.
The limitations of the study should be placed inside the discussion as the last paragraph or before the conclusion of the study.
Also, please revise the references and format of the manuscript according to the instruction for authors.
Author Response
Than you for your insightful comments and suggestions. We have carefully addressed each point, and the corresponding revisions are marked in track changes in the resubmitted manuscript.
The suggestion 1: Please provide a complete description of all abbreviations at the first presentation, including the software description. E.g. IBM SPSS, please provide city of development and full description.
The answer 1: Thank you for pointing this out! A few abbreviations were not given at the first presentation and some of them were not included at the end of the manuscript. Now, the list of abbreviations has been expanded, with 11 additional abbreviations now included in the 'Abbreviations' section (DMPK, PSQ, WISC, AASM, OSAS, MV, CNS, HIE, PCR, ASD, VSD). Furthermore, we have provided the descriptions of some abbreviations upon their first mention in the text (DMPK, ODI, PCR, ASD, VSD, MRI). We hope the section on abbreviations is now complete.
Additionally, the software description has been included (IBM Corp. Released 2020. IBM SPSS Statistics for Windows, Version 27.0. Armonk, NY: IBM Corp) in lines 187-188.
The suggestion 2: How were the variables distributed? Provide the Spearman correlation graphs as supplementary material for data validation.
The answer 2: Thank you for your valuable feedback. We applied tests for both normally and non-normally distributed numerical variables. The same approach was used for correlation, with both Pearson’s and Spearman’s tests employed as needed. Specifically, the Point-Biserial correlation was used when at least one variable was dichotomous. To clarify the data, we reported the values of Pearson’s R, Spearman’s Rho, and Point-Biserial R when p < 0.05, not only to indicate the test used, but also to highlight the strength of the correlation.
Additionally, we included a few graphs for data validation. Since the vast majority of variables did not show significant correlations, we chose to add graphs illustrating positive correlations (AHI to ODI, PtcCO2 to ODI). Although we did not find a correlation between PtcCO2 and AHI, we included this specific graph due to the relevance of the potential correlation between these variables.
Finally, we clarified the relationship between brain atrophy detected by MRI and the oxygen desaturation index. In addition to reporting the value of Welch’s t-statistic, we included bar graphs (now Figure 2) for data validation, as we felt bar graphs were more suitable for a schematic presentation in this particular case.
The suggestion 3: The limitations of the study should be placed inside the discussion as the last paragraph or before the conclusion of the study.
The answer 3: As you suggested, the paragraph discussing the limitations of the study has been placed at the end of the Discussion.
The suggestion 4: Also, please revise the references and format of the manuscript according to the instruction for authors.
The answer 4: The references have been revised, and DOIs have been added for each citation. Tables 1 and 2 have been corrected and adjusted to match the manuscript format. We hope the final format now meets the journal's requirements.
Round 2
Reviewer 1 Report
Comments and Suggestions for Authors
Please revise some details in the Reference section
Author Response
The suggestion: Please revise some details in the Reference section.
The answer: As you suggested, we reviewed all the references and made the following changes: the name/abbreviation of the journal is now in italics, the time of publication is expressed as a year (with the month omitted), and the DOI is formatted as ‘https://doi.org/’. We have corrected the formatting of the volume and year of publication in the references.
We hope these adjustments make the Reference section more suitable for the journal's requirements. the Reference section more suitable to the journal’s requirements.
Reviewer 2 Report
Comments and Suggestions for Authors
The paper is improved, although it has significant limitations in terms of novelty and sample size.
Author Response
The suggestion: The paper is improved, although it has significant limitations in terms of novelty and sample size.
The answer: Thank you for your valuable comments – they helped us improve the paper. However, the paragraph about study limitations within the Discussion section highlights the downsides and limitations of the study. The main limiting factors in terms of sample size were the unwillingness of some parents to participate (as mentioned in the text) and the rare prevalence of MD1 itself. The fact that we reported the experience of two centers (although the number of patients was not equally distributed) suggests that we were aware of these limitations. Nonetheless, we emphasized the need for future multicentric studies to secure more reliable large-scale results.
Additionally, we revised the sentence regarding genetic testing across different time periods. The original version stated:
The retrospective design resulted in the utilization of two distinct techniques for genetic analysis (Repeat-primed PCR and small-pool PCR) over various time periods, although it didn’t affect the reliability of genotype-phenotype correlations.
To improve clarity and precision, we have simplified the sentence to:
The retrospective design resulted in genetic analysis over various time periods, although it didn’t affect the reliability of genotype-phenotype correlations.
Although not essential, we believe this contributes to a slight improvement in the paper's precision.